# Unlocking the Secrets of Acute Coronary Syndromes Using Intravascular Imaging: From Pathophysiology to Improving Outcomes

**DOI:** 10.3390/jcm13237087

**Published:** 2024-11-23

**Authors:** Anastasios Apostolos, Antonios Karanasos, Nikolaos Ktenopoulos, Sotirios Tsalamandris, Panayotis K. Vlachakis, Ioannis Kachrimanidis, Ioannis Skalidis, Marios Sagris, Leonidas Koliastasis, Maria Drakopoulou, Andreas Synetos, Konstantinos Tsioufis, Konstantinos Toutouzas

**Affiliations:** 1First Department of Cardiology, Medical School, National and Kapodistrian University of Athens, “Hippokration” General Hospital of Athens, 11528 Athens, Greece; anastasisapostolos@gmail.com (A.A.); nikosktenop@gmail.com (N.K.); stsalamandris@hotmail.com (S.T.); vlachakispanag@gmail.com (P.K.V.); iskachrimanidis@gmail.com (I.K.); masagris1919@gmail.com (M.S.); lkoliastasis@gmail.com (L.K.); mdrakopoulou@hotmail.com (M.D.); synetos@yahoo.com (A.S.); ktsioufis@gmail.com (K.T.); 2Department of Cardiology, Faculty of Medicine, University of Patras, University Hospital of Patras, 26504 Patras, Greece; akaranasos@hotmail.com; 3Department of Cardiology, Lausanne University Hospital and University of Lausanne, 1011 Lausanne, Switzerland; skalidis7@gmail.com

**Keywords:** intravascular imaging, intravascular ultrasound, optical coherence tomography, STEMI, NSTEMI, review

## Abstract

Acute coronary syndrome (ACS) represents the most severe manifestation of coronary artery disease. Intravascular imaging, both intravascular ultrasound (IVUS) and optical coherence tomography (OCT), have played crucial roles for the impressive reduction in mortality of ACS. Intravascular imaging is useful for the detection of atherosclerotic mechanism (plaque rupture, calcified nodules, or plaque erosions) and for the evaluation of nonatherosclerotic and nonobstructive types of ACS. In addition, IVUS and OCT play a crucial role in the optimization of the PCI. The aim of the current review is to present the role of intravascular imaging in identifying the mechanisms of ACS and its prognostic role in future events, to review the current guidelines suggesting intravascular imaging use in ACS, to summarize its role in PCI in patients with ACS, and to compare IVUS and OCT.

## 1. Introduction

Acute coronary syndrome (ACS), presenting as either ST-elevation myocardial infarction (STEMI) or non-ST-elevation acute coronary syndrome (NSTE-ACS), represents the most severe manifestation of coronary artery disease (CAD) [1]. It constitutes a significant public health challenge, with over one million hospitalizations annually in the United States and a staggering global death toll of 1.8 million individuals per year [2,3]. Coronary artery imaging has emerged as a critical tool in this landscape, enabling precise evaluation of plaque morphology, lesion characteristics, and extent of disease. Enhanced imaging techniques, such as intravascular ultrasound (IVUS) and optical coherence tomography (OCT), are essential for optimizing patient outcomes by guiding percutaneous coronary intervention (PCI) strategies.

The remarkable progress in comprehending the underlying pathology of ACS and the advancements made in interventional procedures, biomaterials utilized during percutaneous coronary intervention (PCI), and pharmacotherapy have significantly contributed to a rapid decline in the mortality associated with ACS over the past decade [4]. These developments have not only improved our ability to diagnose and treat ACS but have also ensured a more favorable outcome for patients suffering from this condition [5]. 

Intravascular imaging in the acute setting remains challenging, and the cost for using intravascular imaging cannot be overlooked. However, both IVUS and OCT have played a crucial role for this impressive reduction, acting in two different ways: first, for the assessment of plaque morphology and characteristics and second, for the optimization of stent implantation [6,7]. The aim of this review is to thoroughly present how intravascular imaging modalities may improve the management and optimize outcomes in patients with ACS. 

## 2. Intravascular Imaging Modalities

Intravascular ultrasound (IVUS) and optical coherence tomography (OCT) are two invasive diagnostic modalities used for in vivo visualization of atherosclerotic plaque morphology. IVUS utilizes high-frequency ultrasound to acquire cross-sectional images of the arterial wall, while OCT uses infrared light [8,9]. Compared with IVUS, OCT has a better axial resolution of approximately 15 μm, allowing better visualization of the superficial structures of the arterial wall, at a cost of reduced penetration with the arterial wall (1–2 mm versus 5–6 mm). The deeper penetration of IVUS allows an in-depth evaluation of the arterial wall layers and the assessment of vascular remodeling, at the cost of less detailed visualization of the superficial structures. Moreover, due to the scattering of the red blood cells, OCT requires the preparation of a blood-free environment, which can be achieved by contrast flashing during image acquisition [10,11,12]. The main characteristics of each imaging modality are presented in Table 1 and the Graphical Abstract.

## 3. Identifying the Atherosclerotic Mechanisms of ACS

Pathological studies have demonstrated that the main pathomechanism of ACS is coronary thrombosis, which is the result of one of three conditions: plaque rupture, plaque erosion, and superficial calcified nodule. In these studies, plaque rupture was identified as the cause in more than 60% of the cases, plaque erosion in about 25%, and calcified nodules in the rest of the cases [13,14]. 

### 3.1. Plaque Rupture

Atherosclerotic plaques are composed of a central necrotic core, encased by a fibrous cap. Histological studies have identified fibrous cap discontinuity and cavity creation as a main mechanism for coronary thrombosis in sudden cardiac death victims. The rupture of this fibrous cap exposes the underlying necrotic core to the bloodstream, leading to the release of pro-thrombogenic substances and subsequently to the formation of a thrombus [15]. The release of thrombogenic contents, as a result of the above described mechanism, was the first proposed explanation for the pathophysiology of ACS and has undergone extensive examination, solidifying its importance in comprehending the condition [13]. 

Post-mortem studies were the first studies that tried to identify plaques prone to trigger thrombosis by exploring morphological similarities of atherosclerotic plaques with plaques associated with thrombus. Plaques bearing morphological resemblances to ruptured plaques have been viewed as the precursors of ACS with a high probability for rupture and thrombosis. These plaques have been dubbed “vulnerable”, and their in vivo detection has garnered significant attention and extensive research as timely identification and treatment of such plaques has been suggested as a potential strategy for the prevention of ACS [16,17,18,19]. 

Intravascular imaging modalities have provided useful insights into the in vivo behavior of these plaques. The first modality that was widely employed for these purposes was intravascular ultrasound with radiofrequency analysis. Several companies have developed their algorithms for plaque characterization based on radiofrequency analysis, with most data including prospective studies derived from IVUS with virtual histology (IVUS-VH). By this modality, the various tissue components can be differentiated and coded in different colors, with dense calcium being represented by white, the necrotic core represented by red, fibro-fatty tissue represented by light green, and fibrous tissue represented by dark green [20,21]. The main drawback of this modality is that the images are a result of IVUS signal post-processing and algorithm training in post-mortem samples, with suboptimal plaque characterization in several cases, especially in the presence of calcium [22]. OCT, on the other hand, can visualize the majority of coronary plaque characteristics, including micro-structures such as cholesterol crystals, macrophages, or microvessels [19].

Thereby, scientific knowledge regarding the in vivo pathomechanisms of ACS has expanded. Thin-cap fibroatheromas (TCFAs) have been found to be primarily situated in the proximal segments of coronary arteries [23,24,25,26], similarly to what has been shown for culprit ruptured plaques in ACS [27]. Plaque morphology by OCT has been associated with the clinical syndrome [28]. Furthermore, it has been demonstrated that the incidence of TCFA detected by OCT is higher in patients with ACS compared with those with stable disease [29,30]. Differences in the morphological characteristics of ruptured plaques give rise to disparities in their clinical presentation. The ruptured plaques associated with STEMI present a greater extent of cap disruption and a smaller minimal lumen area in comparison with those associated with NSTEMI [17]. Compared with ruptured plaques in asymptomatic CAD, ruptured plaques of NSTEACS had smaller luminal dimensions by OCT [31]. Moreover, in patients undergoing thrombolysis, plaque rupture and TCFA morphology has been associated with reduced efficacy of thrombolytic therapy [32].

### 3.2. Plaque Erosion

Despite the established connection of TCFA with plaque rupture and the central role of rupture in ACS pathogenesis, the fibrous cap remains intact in a considerable proportion of patients diagnosed with ACS. This observation suggests that the underlying pathophysiology of ACS may also involve additional mechanisms. Plaque erosion refers to the formation of a thrombus in a region of endothelial denudation, adjacent to a plaque, without any breach in the fibrous cap covering the plaque. This mechanism of thrombogenesis operates in parallel to the rupture of the fibrous cap [33]. Pathologically, it is characterized by the lack of surface endothelium and is formatted on a lesion with thick intima. Rarely, an intact fibrous cap in an ACS setting could be associated with the presence of calcified nodules, which act as a substrate for thrombus formation [34]. 

The only acceptable diagnostic in vivo method for identifying plaque erosion is OCT [35]. Although OCT lacks the necessary resolution to discern endothelial denudation that requires a resolution lower than 10 μm, various definitions have been suggested for plaque erosion by OCT. One of these classifies plaque erosion as “definite” when the plaque consists mainly of fibrous tissue, exhibits the presence of a white luminal thrombus, and maintains an intact fibrous cap. In contrast, an “indefinite” plaque erosion is identified by the lack of a noticeable luminal thrombus or the occurrence of an irregular surface, or the presence of a thrombus without a discernible underlying plaque and superficial lipid or calcium deposits [35]. In contrast with ruptured plaques, eroded plaques present a thicker fibrous cap and smaller lipid burden in OCT imaging [36]. Female sex, lower age, current smoking, isolated lesions, and larger coronary artery diameter are usually the phenotypical characteristics of patients with eroded plaques [36,37,38]. 

The distinction between ruptured and eroded plaques may have a prognostic role. Plaque rupture in patients with ACS has been correlated with a less favorable prognosis and an increased occurrence of cardiac death, nonfatal myocardial infarction, unstable angina, and target lesion revascularization [39]. These results were confirmed by Yonetsu et al.; they investigated 318 ACS patients who underwent OCT and found that an intact fibrous cap, as in plaque erosion, was associated with a better long-term prognosis [40]. Identification of plaque erosion may have not only a prognostic role but also therapeutic implications [41]. Given the histological background knowledge that thrombi associated with eroded plaques exhibit high platelet concentrations and that patients diagnosed with ACS and plaque erosion tend to have a more favorable prognosis, it is plausible that a less invasive therapeutic approach may be effective. In the prospective, proof-of-concept EROSION study, sixty ACS patients with plaque erosion diagnosed by OCT, who had residual diameter stenosis less than 70% and thrombolysis in myocardial infarction (TIMI) flow grade 3 on angiography, were treated with antithrombotic therapy alone without stent implantation. During repeat coronary angiography and OCT evaluation in thirty days, forty seven (78%) met the primary endpoint of >50% reduction in thrombus volume. These findings were maintained through one-year follow-up, and 92.5% of patients were free from MACEs (major adverse cardiovascular events) [42]. Thus, antithrombotic therapy alone was safe and effective in the management of plaque erosion in ACS patients; nevertheless, further larger, randomized controlled studies are required for the validation of EROSION results.

### 3.3. Calcific Nodules

Calcific nodules are described as intraluminal protrusion of calcific elements, with or without disruption of the intimal fibrous layer. Both intravascular imaging techniques can be utilized for the assessment of these nodules. Although initial descriptions of this histopathological entity were made using IVUS, it has limitations in terms of accurately evaluating the nature and composition of these nodules [24,25,43,44].

Moreover, OCT has been a reliable method for detecting and evaluating calcified nodules, being superior compared with conventional IVUS regarding thrombus detection, calcium localization, and plaque continuation assessment [45,46]. Eruptive calcified nodules have been proposed as a risk factor for target vessel failure and a need for repeat revascularization [47]. Determining the nature of calcific ACS lesions through intracoronary imaging can aid in the choice of additional treatment options (pre-dilation, cutting balloons, atherectomy, laser therapy, or lithotripsy).

## 4. Assessment of the Risk of Future Events in ACS Patients

Intravascular imaging and identification of the exact pathophysiological mechanism may have a prognostic role in predicting future events. Prospective studies with IVUS have shown that a combination of high plaque burden and a low residual lumen area is a strong prognostic factor for MACE [48,49]. Newer techniques, like IVUS with virtual histology (IVUS-VH) and near-infrared spectroscopy (NIRS), can identify high-risk features of atherosclerotic plaque, which are associated with MACE at long-term follow-up [48,49,50]. Despite this well-validated association, a wide application of invasive imaging as a risk stratification tool seems unlikely due to the low positive predictive value of this approach suggesting a high number needed to treat, thus hampering attempts for local treatment as a tool for future event prevention [48,49,50]. However, the recently published PREVENT trial challenges the idea that prophylactic stenting of vulnerable plaques is unlikely to find its way in clinical practice [51]. In this multicenter, open-label, randomized-controlled study comparing PCI versus optimal medical treatment in hemodynamically nonsignificant, vulnerable plaques detected by intracoronary imaging, preventive PCI reduced major adverse cardiac events arising from high-risk vulnerable plaques, compared with optimal medical therapy alone. This is the first trial showing the possibility of developing therapeutic strategies based on the detection of high-risk plaque morphology. These findings support the prognostic relevance of the morphological evaluation of nonculprit lesions of ACS patients. The use of intravascular imaging for the detection of high-risk features of nonculprit lesions in patients with ACS might be the next step in the management of patients with ACS and multivessel disease. While an anatomical assessment of lesion severity by coronary angiography is likely the most important criterion to guide lesion selection, intravascular imaging may have an important role in improving risk stratification [52,53]. Interestingly, in the COMPLETE trial, about half of the patients that underwent OCT had at least one nonobstructive, nonculprit lesion with vulnerable plaque morphology [54]. The results of ongoing trials might shed further light on this unresolved issue. The COMBINE-INTERVENE trial is a multicenter, randomized trial exploring whether a PCI revascularization strategy based on combined FFR and OCT assessment is superior to a PCI exclusively based on FFR in patients with multivessel disease presenting with ACS or CCS with any presentation (clinicaltrials.gov: NCT05333068). The Interventional Strategy for Nonculprit Lesions With Major Vulnerability Criteria Identified by Optical Coherence Tomography in Patients With Acute Coronary Syndrome (INTERCLIMA) is a multicenter, randomized trial comparing OCT-based versus physiology-guided treatment of intermediate nonculprit lesions in patients undergoing coronary angiography for ACS (clinicaltrials.gov: NCT05333068). The eagerly awaited results of these studies may shift the paradigm of nonculprit lesion management to a strategy that focuses on the assessment of nonculprit plaque morphology to guide local treatment.

## 5. Nonobstructive Forms of ACS

Myocardial infarction with nonobstructive coronary artery disease (MINOCA) is described as the clinical condition in which patients present with symptoms and electrocardiographic changes consistent with myocardial infarction, but they have angiographically normal or only minimally obstructive (≤50% stenosis) coronary arteries. MINOCA comprises a heterogeneous group of phenotypes, including myocarditis, coronary artery spasm, coronary thromboembolism, spontaneous coronary artery dissection, and Takotsubo syndrome.

Despite the normal angiographic findings, the presence of underlying atherosclerotic disease cannot be ruled out in MINOCA. The underdiagnosis of atherosclerosis in MINOCA may be due to the limitations of invasive coronary angiography, which primarily assesses the larger epicardial coronary arteries and may not identify disease in the microvasculature or other coronary artery segments. In a recent study by Zeng et al., the pathophysiology of MINOCA was explored using OCT. The results showed that approximately half of the patients with MINOCA had an underlying atherosclerotic substrate. Out of the 99 patients studied, 52.1% had either plaque erosion (33.7%), plaque rupture (17.4%), or a calcified nodule (1.1%). A small proportion of patients were diagnosed with SCAD (4.2%) or coronary spasm (4.7%), while a significant number of patients had an unknown cause of MINOCA [55]. In a study conducted by Reynolds et al., the diagnostic utility of OCT and CMR imaging was evaluated in female patients with MINOCA. The results indicated that a significant proportion of patients (65%) had an ischemic mechanism underlying their MINOCA, as determined by the presence of atherosclerotic plaque disruption, as diagnosed by OCT. Moreover, a study that evaluated the use of IVUS in patients presenting with chest pain without obstructive CAD demonstrated that mild to moderate forms of atherosclerosis were present in about 80% of patients who had normal findings on coronary angiography [56]. These findings highlight the importance of utilizing advanced imaging techniques in the investigation of MINOCA, particularly in cases where atherosclerotic substrate could be the underlying cause. Therefore, this information plays a crucial role in guiding the management and treatment of patients that angiographically are characterized as MINOCA cases as a number of them are associated with atherosclerosis [57]. Figure 1 shows the use of intravascular imaging for culprit lesion identification in an ACS patient.

## 6. Nonatherosclerotic Causes of ACS

Intravascular imaging has been shown be a valuable tool to distinguish nonatherosclerotic clinical conditions that can also cause ACS. Spontaneous coronary artery dissection (SCAD) is a nontraumatic separation of the coronary artery vessel wall that can underly an ACS [58]. While traditional coronary angiography may be inadequate in several circumstances for SCAD diagnosis, the use of intravascular imaging has shown promise. In a study conducted by Alfonso and colleagues, SCAD was confirmed using OCT in 11 out of 17 patients diagnosed by traditional angiography with SCAD [59]. In the remaining six patients, OCT was able to rule out the presence of SCAD by revealing critical atherosclerosis, calcified lesions, or thrombus mimicking SCAD.

Both imaging modalities, IVUS and OCT, can diagnose SCAD as the underline cause of an ACS, but each of them has advantages and limitations. The superiority of OCT in the diagnosis of SCAD is attributed to its superior spatial resolution and ability to identify intramural hematoma, endothelial tears, or entry sites of dissection [60]. The main concern with the use of OCT in the diagnostic process of SCAD is the possibility of progression of false lumen due to the contrast injection during imaging acquisition, something that has not been confirmed thus far [61]. The utilization of IVUS is favored in scenarios where there is proof of false lumen, as well as in vessels with small diameter and complex path, where the imaging instrument could potentially cause blockage. The imaging capability of IVUS, including its penetration depth, can be advantageous in cases of proximal vessel dissections where the false lumen expands beyond the external elastic lamina, leading to increased vessel dimensions [62].

Moreover, OCT can also be useful for the management of SCAD as it can identify the site of the intimal tear and determine with high precision the appropriate vessel segment for stenting and also guide the selection of optimal stent diameter for cases that require PCI, such as those with proximal vessel dissection, ongoing ischemia, or reduced/no coronary flow [63]. In most cases, however, a conservative strategy is appropriate due to the autonomous healing process of the vessel wall and the natural history of SCAD with spontaneous healing in most cases [64].

Similarly to spontaneous dissection, coronary spasm is another rare cause of acute coronary syndromes. OCT in this setting has demonstrated a variety of morphological changes in the coronary artery comprising intimal notches and transient separation of the coronary artery wall layers, which is sometimes accompanied by micro-thrombosis [65]. Finally, recanalized thrombus can be identified by OCT appearing as multiple intraluminal tissue rims, resembling a “honeycomb” [66].

Regarding coronary embolism as a cause of ACS, several conditions predispose to it, including thrombophilia, atrial fibrillation, valvular disease, patent foramen ovale, infective endocarditis, and nonbacterial thrombotic endocarditis. Angiography together with a high index of suspicion in patients with risk factors is key in the diagnosis of coronary embolism. Angiographic features, including heavy thrombus burden, abrupt occlusion, or involvement of multiple coronary territories, can provide clues to an embolic phenomenon. The unaffected coronary vessels might appear normal without significant atherosclerotic disease, and the lack of collaterals to the occluded territory might suggest embolism. The absence of plaque rupture or erosion on OCT following thrombectomy in a vessel with minimal atherosclerosis aids in diagnosing coronary embolus. Thromboembolism can also occur in the acute phase of ACS from plaque rupture to the downstream coronary territory (Figure 2). Additional imaging modalities might be necessary for further assessment, such as transesophageal echocardiography to evaluate the left atrial appendage or transthoracic echocardiography with bubble study to identify atrial shunts and determine the appropriate management [67].

## 7. Stent Failure

Despite the radical advancements of PCI and interventional cardiology, stent failure remains a rare but devastating adverse event after PCI and may also cause ACS. Noteworthily, patients with ACS due to stent thrombosis have worse prognosis [68]. Several angiographic and procedural characteristics have been identified as predictors of stent failure, such as lesion length > 20 mm, diameter < 3 mm, ostial location, bifurcation PCI, vein graft PCI, severe calcification, underexpansion, stent fracture, PCI with bare metal stent, stenoses proximal and distal to stent, major dissection, and multiple stent layers [69,70]. Pathomechanisms of stent failure are multifactorial and involve both biological (patient-related) and mechanical (stent-related) factors. Exaggerated neointimal tissue proliferation or hyperplasia, which naturally occurs as a healing response to coronary arterial wall damage during PCI; neoatherosclerosis; and vascular toxicity have been identified as primary biologic mechanisms for stent failure [69,71]. The mechanical factor associated with stent failure is mainly stent underexpansion [69]. The investigation of both mechanical and biological mechanisms, as well as the diagnosis of the exact mechanism, is mainly based on intravascular imaging. This is crucial as it may affect the treatment strategy [72]. 

## 8. Intravascular Imaging Guided PCI

Intracoronary imaging by IVUS or OCT provides a superior understanding of coronary anatomy in comparison with angiography-guided PCI. This is achieved through the high-resolution cross-sectional images that offer detailed information about the structural characteristics of lesions and vessels. Subsequently, the limitations of coronary angiography, which relies on two-dimensional projections to define the coronary artery lumen, are overcome by intracoronary imaging. Thus, by providing more comprehensive anatomical information during the procedure, intracoronary imaging enables the optimization of stent sizing and placement, identification of complications, and improvement of short- and long-term outcomes following the procedure. Recently, the RENOVATE-COMPLEX-PCI trial [73] showed that intravascular imaging PCI was associated with a lower risk of a composite of cardiovascular death, myocardial infarction, or clinically driven target-vessel revascularization compared with angiography-guided PCI. Although this randomized-controlled trial included patients with both CCS and ACS, acute patients were the majority (*N* = 832, 50.8%). However, the total results were driven by the patients with CCS, where intravascular-guided PCI showed a clear benefit (5.0 vs. 10.4, HR: 0.46, 95% CI: 0.27–0.80). The subgroup analysis included only patients with ACS did not reveal any difference between the two arms (10.4 vs. 14.6, HR: 0.74, 95% CI: 0.48–1.15). In the experimental arm, IVUS was utilized in about two thirds of the randomized patients, while OCT was utilized in the rest [74].

### 8.1. IVUS-Guided PCI

Multiple studies, including observational studies, randomized-controlled trials, and meta-analyses, have demonstrated favorable outcomes with IVUS-guided PCI compared with conventional angiography-guided PCI. The results apply to both ACS and CCS cases, as well as in complex and noncomplex PCI scenarios. The early studies in the application of IVUS as a guiding modality for primary PCI demonstrated negative outcomes, leading the recommendation to avoid its routine use. However, the limitations of these studies, including their observational design, the utilization of outdated technology, and the inadequate experience with the technique, should be considered when evaluating these results [75,76,77]. Subsequent studies have challenged the negative outcomes previously reported, supporting the safety and efficacy of IVUS-guided PCI in ACS patients [78]. 

A head-to-head comparison of IVUS-guided and standard-of-care approaches in STEMI patients was performed through a randomized controlled trial led by Wang et al. Patients who were deemed low risk based on IVUS results received conservative pharmaceutical treatment alone, while those considered high risk underwent stent implantation [79]. Recently, a systematic review and meta-analysis conducted by Groenland et al. evaluated the use of IVUS in ACS cases undergoing PCI. The study included data from nine studies and a total of 838,902 patients (796,953 angiography-guided PCI, 41,949 IVUS-guided PCI). The results demonstrated a favorable outcome for IVUS-guided PCI, with a significant reduction in all-cause mortality (0.70), major adverse cardiovascular events (MACE) (0.86), and target vessel revascularization (TVR) (0.83) [80]. Of significance, a recent network meta-analysis and meta-regression analysis revealed that there is evidence to suggest that using IVUS as compared with fractional flow reserve (FFR) may lead to a reduction in subsequent MI in patients with ACS [81]. The most important body of evidence supporting the use of intravascular imaging for PCI guidance in ACS comes from the IVUS-ACS trial, a prospective, multicenter, randomized controlled trial designed to investigate the impact of IVUS guidance versus angiography guidance in DES implantation on the incidence of target vessel failure (TVF) at 12 months among ACS patients and to compare the effect of ticagrelor monotherapy versus ticagrelor plus aspirin on the risk of clinically relevant bleeding and MACCE in the first 12 months after PCI in ACS patients undergoing DES implantation guided by either IVUS or angiography [82]. It showed that in patients with ACS, IVUS-guided implantation of drug-eluting stents was associated with a lower 1-year rate of the composite outcome, which included cardiac death, target vessel MI, or clinically driven revascularization, compared with angiographic guidance. The IVUS-ACS and ULTIMATE-DAPT study distinguish themselves from other studies on IVUS-guided PCI by exploring the correlation between IVUS guidance and the shortening of the duration of DAPT, which has recently been deemed a favorable approach [83,84,85,86,87].

Ongoing trials, such as the iSTEMI trial (NCT04775914), the IVUS-CHIP trial (NCT04854070), and the SPECTRUM study (NCT05007535), will support with greater evidence the current literature. Despite the beneficial role of IVUS in PCI guiding for ACS, its utilization in current clinical practice is limited; recent data from the United Kingdom showed than fewer than one in seven patients with ACS underwent IVUS-guided PCI [88]. 

### 8.2. OCT-Guided PCI

Optical coherence tomography, a relatively recent and less-investigated technology, is gaining widespread adoption in clinical practice. The improvement in technology and the utilization of second-generation frequency domain (FD) OCT systems have been instrumental in mitigating the risk of arterial occlusion and optimizing stent selection [89,90]. FD-OCT is considered to have a comparable safety profile to IVUS, with minimal and infrequent adverse events associated with invasive imaging [91]. The existing literature suggests that OCT is comparable to IVUS in terms of procedural endpoints like stent expansion and MLA and better than coronary angiography [92,93,94,95,96]. Two recent trials have compared OCT-guided PCI with angiography-guided PCI. More specifically, the OCTOBER trial was a European, multicenter, RCT comparing routine OCT guidance in PCI of bifurcations with angiographic guidance with respect to MACE in a median two-year follow-up. OCT-guided PCI was associated with significantly fewer adverse events (10.1% vs. 14.1%, *p* = 0.035) [97]. The positive results of OCTOBER were not corroborated by ILUMIEN IV. Compared with OCTOBER, the ILUMIEN IV study was not exclusively focused on bifurcations but included patients with diabetes mellitus or those undergoing PCI of complex lesions. Although OCT guidance resulted in a larger minimum stent area than angiography guidance, no significant difference in the incidence of TVF was observed [98]. 

Limited data specifically relating to patients with ACS are available. A propensity matched analysis from the International FORMIDABLE-CARDIOGROUP IV and USZ Registry compared OCT-guided vs. angiography-guided PCI in an ACS setting. A total of 540 patients were included and divided into two groups. The OCT guidance led to a significant decrease in stent implantation and numerically lower rates of MACE, TVR, and stent thrombosis [99]. 

The EROSION III study was a multicenter, randomized controlled trial that compared the OCT-guided reperfusion approach to the conventional angiographic guidance in 226 patients with STEMI and early infarct artery patency. The application of OCT led to a reduction of 15% in stent implantation. The incidence of cardiovascular and cerebral events during the first-year follow-up did not significantly differ between the two groups. Nevertheless, the minimally invasive approach facilitated by OCT merits further exploration for potential future use [100]. A recent case-controlled study suggested that the improved results associated with OCT-guided procedures might be due to better stent expansion [101]. Additional prognostic information might be provided by FFR computed from OCT images [102]. The TACTICS registry, an investigator-initiated, prospective, multicenter, observational study to be conducted at 21 hospitals in Japan, will try to shed light on numerous unanswered questions about OCT-guided PCI in ACS, such as the identification of ACS causes and clinical outcomes in a two-year follow-up [103].

### 8.3. IVUS-Versus OCT-Guided PCI

The choice between IVUS and OCT to complement coronary angiography remains controversial. Historically, IVUS was deemed more appropriate in most cases; however, advancements in OCT have narrowed the differences between the two modalities. Recent randomized controlled trials have demonstrated that OCT is not inferior to IVUS in terms of periprocedural and clinical endpoints [92,104,105,106]. These results were further confirmed by the recent OCTIVUS trial, in which OCT-guided was noninferior to IVUS-guided PCI regarding the incidence of a composite endpoint including cardiac, target vessel-related MI, or TVR at 12 months. The results did not differ between patients with ACS and stable patients [107]. Direct comparison of OCT and IVUS for PCI guidance in ACS has not been performed. Considering the superior ability of OCT to detect thrombus, it may be hypothesized that it is superior to IVUS for guidance in ACS. The OPINION ACS study intends to examine if FD-OCT is superior to IVUS in patients with ACS undergoing PCI [108]. Overall and pending further validation, considering the results of OPINION and ILUMIEN III, an OCT-guided PCI strategy appears to be noninferior compared with IVUS for both acute and long-term outcomes. The key studies summarizing OCT and IVUS applications in ACS patients are presented in Table 2.

## 9. Intravascular Imaging and Guidelines on ACS

Intravascular imaging, both IVUS and OCT, has gained significant ground in the diagnostic and therapeutic approach to coronary syndromes, both acute and chronic. Thus, recent guidelines have incorporated intravascular imaging as part of the recommended clinical practice [112]. The most recent guidelines of ESC about ACS recommend that intravascular imaging should be utilized in cases of angiographically suspected SCAD, in which there is obstruction but with normal coronary flow. In such cases, IVUS or OCT could play a fundamental diagnostic role by proving the presence of double lumen or intramural hematoma. For giving more accurate options in unclear situations, intravascular imaging would be valuable and in MINOCA, contributing to nondiagnostic findings from conventional angiography, such as plaque erosion, nodule, or rupture. Moreover, intravascular imaging could provide useful information for detecting the culprit lesion as it is not clear in a significant proportion of patients with NSTE-ACS, while in about one out of ten patients, more than one lesion is the culprit. Additionally, current guidelines emphasize the use of intravascular imaging as a tool for guiding and optimizing PCI [112]. In the recent 2024 ESC guidelines on chronic coronary syndrome management, intracoronary imaging guidance by IVUS or OCT is recommended for performing PCI on anatomically complex lesions, such as the left main stem, bifurcations, or long lesions (class I, level A) [113].

## 10. Future Directions

Taking into consideration the individual advantages and disadvantages of IVUS and OCT, hybrid systems combining IVUS and OCT have been proposed as an interesting approach, combining the benefits of each method. Despite several attempts to develop such a novel intravascular imaging catheter and multiple in vitro and in vivo studies in animal subjects, only one hybrid catheter, namely, the Novasight Hybrid™ system by Conavi Medical Inc. (Toronto, ON, Canada), has been studied in humans [114,115,116]. The co-registered acquisition of IVUS and OCT images was safe in 17 patients, with acceptable imaging results both before and after PCI [117]. 

Intracoronary imaging and physiology assessment are both useful for lesion evaluation and PCI optimization, with discrete advantages and disadvantages in lesion selection and procedural guidance. Given the benefits of physiology and imaging, simultaneous utilization of both methods might have further clinical implications. The OCT-based flow ratio, a method that employs computational methods for assessing the fractional flow reserve by anatomical information provided by OCT images, could be used to evaluate the functional significance of coronary stenoses. The Functional Diagnosis of Coronary Stenosis (FUSION) trial (NCT04356027) using the ILUMIEN™ OPTIS™ platform (Abbott Vascular, Santa Clara, CA, USA) is ongoing, aiming to compare the diagnostic performance of virtual flow reserve with conventional FFR. Although this technology would mainly find application in stable CCS, it could also be used for lesion assessment in ACS. 

Considering that OCT is a relatively new imaging modality in clinical practice, further improvements to its technology could be performed. The OCT scan during a single cardiac cycle with full sampling both cross-sectionally and longitudinally might eliminate artifacts due to cardiac motion [118]. Additionally, the micro-OCT with an improved resolution (up to 2 μm) could identify cells (macrophages and endothelial cells) and other structures (cholesterol crystals), providing further basic and clinical information [119], while polarization-sensitive OCT may provide additional information regarding the structural composition of the plaque [120,121]. Moreover, the combined OCT and fluorescence lifetime imaging could provide structural and biochemical details of coronary atherosclerosis simultaneously (NCT04835467) [122]. However, further studies are required in stable patients undergoing elective revascularization, prior to application in acute settings.

Considering the ability of intravascular imaging modalities to detect features of vulnerability even in nonobstructive lesions potentially responsible for future acute thrombotic events, further studies should focus on intravascular imaging-based risk stratification and development of therapeutic strategies of prophylactic stenting. The results of the PREVENT trial, in anticipation of the results of additional trials exploring the potential prognostic benefit of local treatment of vulnerable nonculprit or nonhemodynamically significant lesions in patients with ACS, hold promise for an enhanced prevention of future events. Against this background, intravascular imaging might be combined with noninvasive imaging modalities, like cardiac magnetic resonance, which show promising results in post-ACS inflammation evaluation and in MACE prediction [123].

## 11. Conclusions

Intravascular imaging modalities have an established role for the effective management of ACS, from both a diagnostic and a therapeutic aspect. Despite the scarcity of studies regarding the implementation of IVUS and OCT in the management of ACS, these modalities have demonstrated positive outcomes in terms of optimizing interventions and enhancing clinical outcomes. Further research through ongoing trials will aim to address the remaining uncertainties in the field and better define the role of intravascular imaging in ACS.

## Figures and Tables

**Figure 1 jcm-13-07087-f001:**
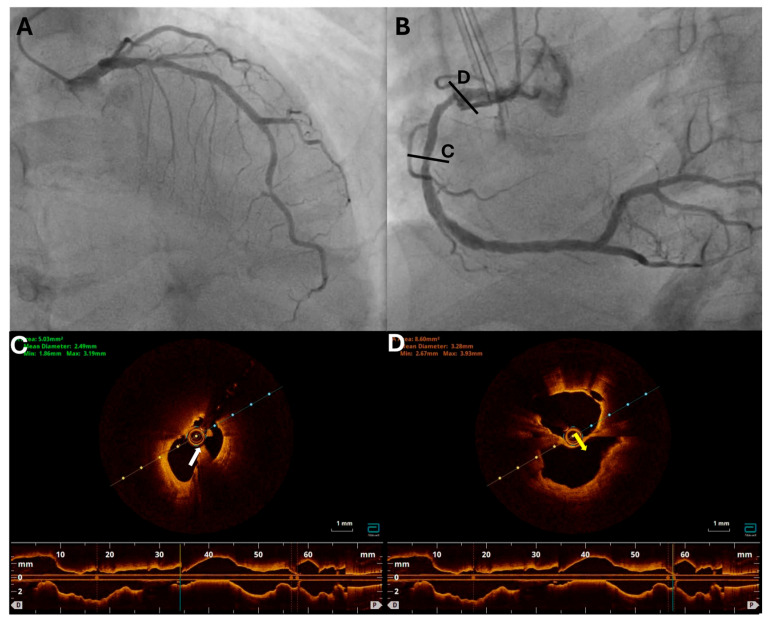
Coronary angiography and OCT images from a 71-year-old male with NSTEMI and echocardiographic evidence of inferior wall hypokinesia. (**A**) Coronary angiography of the left coronary artery showing a mild stenosis of the proximal LAD and a severe stenosis of the distal LAD. (**B**) Coronary angiography of the RCA showing intermediate stenosis with ulcerates appearance in the proximal and middle RCA. OCT imaging of the RCA was performed due to the echocardiographic findings and the suspicious angiographic appearance showing the presence of rupture with thrombus formation in the middle RCA (panel (**C**); white arrow), as well as a rupture with large cavity formation in proc RCA (panel (**D**); yellow arrow), suggesting that RCA was the culprit vessel. LAD, left anterior descending; OCT, optical coherence tomography; NSTEMI, non-ST-elevation myocardial infarction; PCI, percutaneous coronary intervention; RCA, right coronary artery.

**Figure 2 jcm-13-07087-f002:**
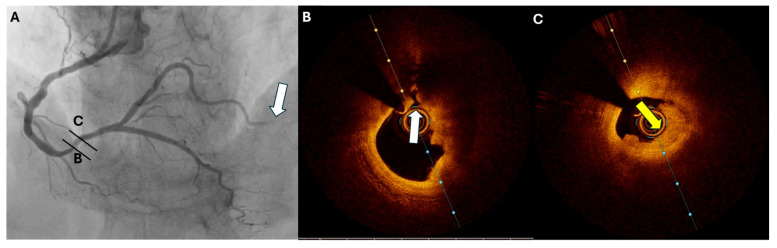
Coronary angiography and OCT images from a 74-year-old male with inferior STEMI. (**A**) Coronary angiography shows a totally occluded PVL artery (arrow) and the presence of a distal RCA lesion. OCT imaging discloses the presence of (**B**) plaque rupture (white arrow) and (**C**) near-occlusive thrombosis (yellow arrow) at the distal RCA lesion. OCT, optical coherence tomography; STEMI, ST-elevation myocardial infarction; PLV, posterior left ventricular; RCA, right coronary artery.

**Table 1 jcm-13-07087-t001:** Different intravascular imaging modalities in acute coronary syndromes.

Modality	Intravascular Ultrasound (IVUS)	Optical Coherence Tomography (OCT)
Resolution	Lower resolution (approximately 100–150 µm) compared with OCT. Suitable for larger structures.	Higher resolution (10–20 µm), providing detailed images of plaque microstructure.
Tissue penetration	Deeper tissue penetration (1–2 mm), allowing better visualization of larger plaques and vessel wall.	Limited penetration depth (5–6 mm), suitable for detailed images of superficial layers.
Plaque characterization	Effective for identifying calcified and fibrotic plaques and measuring plaque burden.	Superior for detecting plaque erosion, thin-cap fibroatheroma, and thrombus.
Blood clearance	Not required; can image through blood.	Requires blood clearance to obtain clear images.
Stent optimization	Good for assessing stent expansion and apposition.	Provides high-resolution images for detailed stent positioning and identification of malapposition or edge dissection.
Speed of imaging	Generally quicker; no need for flushing makes it practical for routine clinical use.	Slightly slower due to need for blood clearance, which may add complexity.
Vessel sizing	Well suited for accurate vessel sizing, especially in larger vessels.	Accurate for vessel sizing but may be limited in larger or highly calcified vessels.
Thrombus Visualization	Limited visualization of thrombus compared with OCT.	Excellent for visualizing intracoronary thrombus, especially in acute coronary syndromes.
Safety and patient tolerance	Lower potential for vessel injury due to lower resolution.	May increase risk of vessel spasm and require additional steps, like saline flush, which can be less tolerated in some patients.
Clinical utility	Widely available and more commonly used in routine PCI procedures.	More specialized use, especially valuable for high-resolution assessment of complex lesions.
Cost and accessibility	Generally more accessible and cost-effective than OCT.	Higher cost and limited availability compared with IVUS.

**Table 2 jcm-13-07087-t002:** Key studies summarizing OCT and IVUS applications in ACS patients.

Study	Year	Study Design	Population	Imaging Modality	Main Findings
ILUMIEN IV [98]	2020	RCT	ACS patients undergoing PCI	OCT vs. angiography	Although there was no reduction in the primary clinical endpoint (target vessel failure), OCT was associated with a reduction in stent thrombosis vs. angiography.
IVUS-ACS and ULTIMATE-DAPT [109]	2018	RCT	ACS and stable CAD patients undergoing PCI	IVUS vs. angiography	IVUS-guided PCI resulted in a significantly lower rate of target vessel failure; notable outcomes in ACS subgroup.
OPINION [110]	2018	RCT	ACS and stable CAD patients	OCT vs. IVUS	OCT-guided PCI was noninferior to IVUS in clinical outcomes for ACS and stable patients; no significant difference in MACE at 1 year.
CLI-OPCI II [111]	2015	Observational	ACS patients undergoing PCI	OCT	OCT-guided PCI showed lower rates of stent malapposition and improved clinical outcomes compared with angiography in ACS patients.
OCTIVUS-ACS [107]	2022	Observational	Complex ACS patients undergoing PCI	OCT and IVUS	Dual OCT and IVUS guidance improved stent expansion and reduced malapposition, particularly beneficial in high-risk ACS cases.

ACS, acute coronary syndrome; IVUS, intravascular ultrasound; MACE, major adverse cardiac event; OCT, optical coherence tomography; PCI, percutaneous coronary interventiοn.

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
