# Peer review of "Unlocking the Secrets of Acute Coronary Syndromes Using Intravascular Imaging: From Pathophysiology to Improving Outcomes"

_jcm, 2024, doi:10.3390/jcm13237087_

Round 1

Reviewer 1 Report

Comments and Suggestions for Authors

The manuscript comprehensively reviews coronary artery imaging, focusing on IVUS, OCT, FFR, and their clinical applications in guiding and optimizing coronary interventions. The authors have cited many relevant studies and guidelines, strengthening the manuscript’s credibility. However, there are several areas where clarity, structure, and grammar could be improved. Below are my specific comments and suggestions:

1. Introduction: The introduction section did not sufficiently present the objectives and aims. Consider providing a brief background on the importance of coronary artery imaging and its impact on clinical practice.

2. Application:

- Consider briefly explaining why this is particularly challenging and how IVUS and OCT can contribute to better outcomes.

- Also, the classification scheme for atherosclerotic lesions is crucial. Consider briefly describing the different categories and their clinical relevance.

- The comparison between plaque erosion and plaque rupture is valuable. Consider summarizing the key findings and their clinical implications.

3. Syntax

- Ensure consistency in the use of terminology and acronyms, such as ensuring IVUS and OCT are defined upon first use.

 4. Conclusion:

- The conclusion should briefly summarize the key findings and potential implications of the reviewed imaging techniques for clinical practice and future research.

Author Response

Dear Editor,

Dear Reviewers,

Thank you for giving us the opportunity to submit a revised draft of the manuscript “Unlocking the Secrets of Acute Coronary Syndromes Using Intravascular Imaging: From Pathophysiology to Improving Outcomes” for publication in the Journal of Clinical Medicine.

We appreciate the time and effort that you and the reviewers dedicated to providing feedback on our manuscript and are grateful for the insightful comments on and valuable improvements to our paper.

We have incorporated most of the suggestions. Those changes are highlighted within the manuscript. Please see below, for a point-by-point response to the comments and concerns.

  1. Introduction: The introduction section did not sufficiently present the objectives and aims. Consider providing a brief background on the importance of coronary artery imaging and its impact on clinical practice.

Answer: We thank the first reviewer for his/her valuable comments. We added the following: Coronary artery imaging has emerged as a critical tool in this landscape, enabling precise evaluation of plaque morphology, lesion characteristics, and extent of disease. Enhanced imaging techniques, such as intravascular ultrasound (IVUS) and optical coherence to-mography (OCT), are essential for optimizing patient outcomes by guiding percutaneous coronary intervention (PCI) strategies.

  1. Application:

- Consider briefly explaining why this is particularly challenging and how IVUS and OCT can contribute to better outcomes.

- Also, the classification scheme for atherosclerotic lesions is crucial. Consider briefly describing the different categories and their clinical relevance.

- The comparison between plaque erosion and plaque rupture is valuable. Consider summarizing the key findings and their clinical implications.

Answer: We thank the first reviewer for his/her comment. We added the following:

Intravascular imaging in acute setting remain challenging and the cost for using intravascular imaging cannot be overlooked.

Despite the established connection of TCFA with plaque rupture and the central role of rupture in ACS pathogenesis, the fibrous cap remains intact in a considerable pro-portion of patients diagnosed with ACS. This observation suggests that the underlying pathophysiology of ACS may also involve additional mechanisms. Plaque erosion refers to the formation of a thrombus in a region of endothelial denudation, adjacent to a plaque, without any breach in the fibrous cap covering the plaque. This mechanism of thrombogenesis operates in parallel to the rupture of the fibrous cap.[31] Pathologically, it is characterized by the lack of surface endothelium and is formatted on a lesion with thick intima. Rarely, intact fibrous cap in ACS setting could be associated with the presence of calcified nodules, which act as substrate for thrombus formation.[32]

In addition, we added a graphical abstract in order to summarize the main findings of our review.

  1. Syntax

- Ensure consistency in the use of terminology and acronyms, such as ensuring IVUS and OCT are defined upon first use.

Answer: We thank the Reviewer for her/his comment. We checked again our manuscript for syntactical errors or typos and we corrected it appropriately.

  1. Conclusion:

- The conclusion should briefly summarize the key findings and potential implications of the reviewed imaging techniques for clinical practice and future research.

Answer: We thank the Editor for/her this valuable comment. We have modified conclusion as follows:

Intravascular imaging modalities have an established role for the effective management of ACS, both from a diagnostic and a therapeutic aspect. Despite the scarcity of studies regarding the implementation of IVUS and OCT in the management of ACS, these modalities have demonstrated positive outcomes in terms of optimizing interventions and enhancing clinical outcomes. Further research through ongoing trials will aim to address the remaining uncertainties in the field and better define the role of intravascular imaging in ACS.

Reviewer 2 Report

Comments and Suggestions for Authors

Thank you to the authors for submitting their manuscript to our journal. This review, titled "Unlocking the Secrets of Acute Coronary Syndromes Using Intravascular Imaging: From Pathophysiology to Improving Outcomes," adequately addresses its subject matter, although it lacks significant novel contributions. To enhance its utility for readers, major revisions are necessary, as outlined below:

1) The current repetition rate stands at 36%; efforts should be made to lower this figure to enhance clarity and conciseness.

2) The authors are encouraged to create a graphical abstract that succinctly summarizes the main messages of the review, facilitating easier comprehension for the readers.

3) It is essential that the introduction specifies that this is a narrative review, providing rationale for choosing this format. Furthermore, the introduction should delve deeper into emerging aspects related to acute coronary syndromes, such as inflammation and malnutrition, which significantly impact prognosis. The work of Trimarchi et al. (PMCID: PMC11508711, DOI: 10.3390/jcm13206059) should be cited in this context.

4) The review should offer a more comprehensive discussion on the use of Optical Coherence Tomography (OCT) in acute coronary syndromes, particularly its influence on therapeutic strategies. Relevant citations include the study by Buonpane et al. (PMCID: PMC11477163, DOI: 10.3390/jcm13195791).

5) It is recommended to include images (OCT and IVUS) for each plaque type discussed (plaque rupture, plaque erosion, calcific nodules) to enhance the visual understanding of the subject matter.

6) Providing summary tables of key studies related to IVUS and OCT in the context of acute coronary syndromes would greatly benefit the review. 

Author Response

Dear Editor,

Dear Reviewers,

Thank you for giving us the opportunity to submit a revised draft of the manuscript “Unlocking the Secrets of Acute Coronary Syndromes Using Intravascular Imaging: From Pathophysiology to Improving Outcomes” for publication in the Journal of Clinical Medicine.

We appreciate the time and effort that you and the reviewers dedicated to providing feedback on our manuscript and are grateful for the insightful comments on and valuable improvements to our paper.

We have incorporated most of the suggestions. Those changes are highlighted within the manuscript. Please see below, for a point-by-point response to the comments and concerns.

1) The current repetition rate stands at 36%; efforts should be made to lower this figure to enhance clarity and conciseness.

Answer: We thank the second reviewer for this valuable comments. We modified the manuscript, aiming to reduce the repetition rate.

2) The authors are encouraged to create a graphical abstract that succinctly summarizes the main messages of the review, facilitating easier comprehension for the readers.

Answer: We thank the second reviewer for his/her comment. Following your recommendation, we prepared and added a graphical abstract summarizing the main findings of our review.

3) It is essential that the introduction specifies that this is a narrative review, providing rationale for choosing this format. Furthermore, the introduction should delve deeper into emerging aspects related to acute coronary syndromes, such as inflammation and malnutrition, which significantly impact prognosis. The work of Trimarchi et al. (PMCID: PMC11508711, DOI: 10.3390/jcm13206059) should be cited in this context.

Answer: We thank the Reviewer for this comment. We added the recommended citation in our reference list.

4) The review should offer a more comprehensive discussion on the use of Optical Coherence Tomography (OCT) in acute coronary syndromes, particularly its influence on therapeutic strategies. Relevant citations include the study by Buonpane et al. (PMCID: PMC11477163, DOI: 10.3390/jcm13195791).

Answer: We thank the Reviewer for this valuable comment. We added the recommended citation in our reference list.

5) It is recommended to include images (OCT and IVUS) for each plaque type discussed (plaque rupture, plaque erosion, calcific nodules) to enhance the visual understanding of the subject matter.

Answer: We thank the Reviewer for this valuable comment. We agree that these additions would be interesting; however, they may be outside the scope of the current review article. Our aim is to focus on the clinical applications of IVUS and OCT in patients presented with Acute Coronary Syndrome, rather than on technical aspects like imaging findings.

6) Providing summary tables of key studies related to IVUS and OCT in the context of acute coronary syndromes would greatly benefit the review. 

Answer: Answer: We thank the Reviewer for his/her this valuable comment. We added the following table.

Table 2: Key studies summarizing OCT and IVUS applications in ACS patients

Study

Year

Study Design

Population

Imaging Modality

Main Findings

ILUMIEN IV[98]

2020

RCT

ACS patients undergoing PCI

OCT vs. Angiography

Although there was no reduction in the primary clinical endpoint (target vessel failure), OCT was associated with a reduction in stent thrombosis vs. angiography.

IVUS-ACS and ULTIMATE-DAPT[109]

2018

RCT

ACS and stable CAD patients undergoing PCI

IVUS vs. Angiography

IVUS-guided PCI resulted in a significantly lower rate of target vessel failure; notable outcomes in ACS subgroup.

OPINION[110]

2018

RCT

ACS and stable CAD patients

OCT vs. IVUS

OCT-guided PCI was non-inferior to IVUS in clinical outcomes for ACS and stable patients; no significant difference in MACE at 1 year.

CLI-OPCI II[111]

2015

Observational

ACS patients undergoing PCI

OCT

OCT-guided PCI showed lower rates of stent malapposition and improved clinical outcomes compared to angiography in ACS patients.

OCTIVUS-ACS[107]

2022

Observational

Complex ACS patients undergoing PCI

OCT and IVUS

Dual OCT and IVUS guidance improved stent expansion and reduced malapposition, particularly beneficial in high-risk ACS cases.

ACS: Acute Coronary Syndrom, IVUS: Intravascular Ultrasound, MACE: Major Adverse Cardiac Events, OCT: Optical Coherence Tomography, PCI: Percutaneous Coronary Interventιοn,

Reviewer 3 Report

Comments and Suggestions for Authors

The authors evaluated the role of intravascular imaging, specifically intravascular ultrasound and optical coherence tomography, in managing acute coronary syndromes. They discussed its significance in identifying atherosclerotic mechanisms, optimizing percutaneous coronary intervention, and assessing the prognostic implications for future events. The review also aimed to summarize current guidelines on the use of intravascular imaging in acute coronary syndromes and compare the effectiveness of intravascular ultrasound and optical coherence tomography.

Congratulations to the authors on this interesting review addressing a highly relevant topic. 

Issues/Suggestions:

Figure 1, panel A: Since the description refers to the left coronary artery and left anterior descending artery, I suggest including the complete coronary angiography. In Figure 2, I believe the description of the angiography is reversed compared to Figure 1.

A case of MINOCA evaluated with intravascular imaging could be beneficial.

I recommend adding a table that assesses the pros and cons of IVUS and OCT imaging evaluations, also considering the limitations and risks of both methods for plaque assessment in the text.

I suggest streamlining the review by including a table summarizing studies that have evaluated OCT- and IVUS-guided PCI strategies.

Currently, integrated multimodal imaging is essential for patients with acute coronary syndromes. Predictors of reinfarction include both plaque burden and type assessed with OCT/IVUS. Additionally, some non-invasive imaging methods, particularly CMR, which evaluate post-ACS inflammation, are crucial for assessing the risk of events during follow-up, especially recurrent myocardial infarction (cite: PMID: 38276932). I recommend expanding the "Future Directions" section to consider the potential integration of invasive and non-invasive methods.

Perform a thorough proofread to correct minor grammatical issues and ensure clarity and readability.

Comments on the Quality of English Language

It is recommended that abbreviations and typos be reviewed carefully.

Author Response

Dear Editor,

Dear Reviewers,

Thank you for giving us the opportunity to submit a revised draft of the manuscript “Unlocking the Secrets of Acute Coronary Syndromes Using Intravascular Imaging: From Pathophysiology to Improving Outcomes” for publication in the Journal of Clinical Medicine.

We appreciate the time and effort that you and the reviewers dedicated to providing feedback on our manuscript and are grateful for the insightful comments on and valuable improvements to our paper.

We have incorporated most of the suggestions. Those changes are highlighted within the manuscript. Please see below, for a point-by-point response to the comments and concerns.

Figure 1, panel A: Since the description refers to the left coronary artery and left anterior descending artery, I suggest including the complete coronary angiography. In Figure 2, I believe the description of the angiography is reversed compared to Figure 1.

Answer: We thank the third reviewer for this right comment.  Indeed, we accidentally reversed the two descriptions, and we corrected in the current version.

A case of MINOCA evaluated with intravascular imaging could be beneficial.

Answer: We thank the Reviewer for the suggestion to include a MINOCA case evaluated with intravascular imaging. While MINOCA is an important clinical entity, it often lacks specific intravascular findings, as these cases typically do not involve obstructive coronary plaques or characteristic morphological changes. Including a MINOCA case may therefore not provide meaningful insights into the plaque assessment and stent optimization aspects that are central to this review. Our focus remains on cases with obstructive lesions where IVUS and OCT findings directly impact clinical decision-making and procedural outcomes.

I recommend adding a table that assesses the pros and cons of IVUS and OCT imaging evaluations, also considering the limitations and risks of both methods for plaque assessment in the text.

Answer: We thank the Reviewer for her/his comment. We added the following table, as you recommended.

Table 1: Different intravascular imaging modalities in acute coronary syndromes

Modality

Intravascular Ultrasound (IVUS)

Optical Coherence Tomography (OCT)

Resolution

Lower resolution (approximately 100-150 µm) compared to OCT. Suitable for larger structures.

Higher resolution (10-20 µm), providing detailed images of plaque microstructure.

Tissue Penetration

Deeper tissue penetration, allowing better visualization of larger plaques and vessel wall.

Limited penetration depth, suitable for detailed images of superficial layers.

Plaque Characterization

Effective for identifying calcified and fibrotic plaques, and measuring plaque burden.

Superior for detecting plaque erosion, thin-cap fibroatheroma and thrombus.

Blood Clearance

Not required; can image through blood.

Requires blood clearance to obtain clear images.

Stent Optimization

Good for assessing stent expansion and apposition.

Provides high-resolution images for detailed stent positioning and identification of malapposition or edge dissection.

Speed of Imaging

Generally quicker; no need for flushing makes it practical for routine clinical use.

Slightly slower due to need for blood clearance, which may add complexity.

Vessel Sizing

Well-suited for accurate vessel sizing, especially in larger vessels.

Accurate for vessel sizing but may be limited in larger or highly calcified vessels.

Thrombus Visualization

Limited visualization of thrombus compared to OCT.

Excellent for visualizing intracoronary thrombus, especially in acute coronary syndromes.

Safety and Patient Tolerance

Lower potential for vessel injury due to lower resolution.

May increase risk of vessel spasm and require additional steps, like saline flush, which can be less tolerated in some patients.

Clinical Utility

Widely available and more commonly used in routine PCI procedures.

More specialized use, especially valuable for high-resolution assessment of complex lesions.

Cost and Accessibility

Generally more accessible and cost-effective than OCT.

Higher cost and limited availability compared to IVUS.

I suggest streamlining the review by including a table summarizing studies that have evaluated OCT- and IVUS-guided PCI strategies.

Answer: We thank the Reviewer for his/her this valuable comment. We added the following table.

Table 2: Key studies summarizing OCT and IVUS applications in ACS patients

Study

Year

Study Design

Population

Imaging Modality

Main Findings

ILUMIEN IV[98]

2020

RCT

ACS patients undergoing PCI

OCT vs. Angiography

Although there was no reduction in the primary clinical endpoint (target vessel failure), OCT was associated with a reduction in stent thrombosis vs. angiography.

IVUS-ACS and ULTIMATE-DAPT[109]

2018

RCT

ACS and stable CAD patients undergoing PCI

IVUS vs. Angiography

IVUS-guided PCI resulted in a significantly lower rate of target vessel failure; notable outcomes in ACS subgroup.

OPINION[110]

2018

RCT

ACS and stable CAD patients

OCT vs. IVUS

OCT-guided PCI was non-inferior to IVUS in clinical outcomes for ACS and stable patients; no significant difference in MACE at 1 year.

CLI-OPCI II[111]

2015

Observational

ACS patients undergoing PCI

OCT

OCT-guided PCI showed lower rates of stent malapposition and improved clinical outcomes compared to angiography in ACS patients.

OCTIVUS-ACS[107]

2022

Observational

Complex ACS patients undergoing PCI

OCT and IVUS

Dual OCT and IVUS guidance improved stent expansion and reduced malapposition, particularly beneficial in high-risk ACS cases.

ACS: Acute Coronary Syndrom, IVUS: Intravascular Ultrasound, MACE: Major Adverse Cardiac Events, OCT: Optical Coherence Tomography, PCI: Percutaneous Coronary Interventιοn,

Currently, integrated multimodal imaging is essential for patients with acute coronary syndromes. Predictors of reinfarction include both plaque burden and type assessed with OCT/IVUS. Additionally, some non-invasive imaging methods, particularly CMR, which evaluate post-ACS inflammation, are crucial for assessing the risk of events during follow-up, especially recurrent myocardial infarction (cite: PMID: 38276932). I recommend expanding the "Future Directions" section to consider the potential integration of invasive and non-invasive methods.

Answer: We thank the Reviewer for this valuable comment. We added the following: Against this background, intravascular imaging might be combined with non-invasive imaging modalities, like cardiac magnetic resonance, which show promising results in post-ACS inflammation evaluation and in MACE prediction and the relevant reference.

Perform a thorough proofread to correct minor grammatical issues and ensure clarity and readability.

Answer: We thank the Reviewer for this valuable comment. Following the recommendation of the Reviewer, we checked the manuscript for minor grammatical issues and made it the appropriate corrections.

Round 2

Reviewer 2 Report

Comments and Suggestions for Authors

I appreciate effort made by the authors in addressing the previous comments. However, the requested citations have not been included in the manuscript references. Please integrate the following references per the previous review and highlight them accordingly in the text:

1) The introduction should explore emerging aspects related to acute coronary syndromes, such as inflammation and malnutrition, which significantly impact prognosis. Please cite the work of Trimarchi et al. (PMCID: PMC11508711, DOI: 10.3390/jcm13206059) in this context.

2) The review should provide a more comprehensive discussion on the use of Optical Coherence Tomography (OCT) in acute coronary syndromes, specifically regarding its influence on therapeutic strategies. Relevant citations include the study by Buonpane et al. (PMCID: PMC11477163, DOI: 10.3390/jcm13195791).

These minor revisions will enhance the quality of the manuscript. Thank you for your attention to these matters.

Author Response

Accidentally, we did not insert the latest version of references, where the recommended citations were included

Reviewer 3 Report

Comments and Suggestions for Authors

Congratulations to the authors as the paper has significantly improved. However, the authors must check the current version of the bibliography, as it is incorrect. Some citations are not included in the bibliography as it stands. Please verify this.

Author Response

Dear Reviewer,

We have updated the reference list and we think that now is alright.

Round 3

Reviewer 3 Report

Comments and Suggestions for Authors

The bibliography is not yet accurate. There are currently 123 citations within the text, but only 115 are listed in the bibliography. I encourage the authors to correct this discrepancy.

Author Response

Sorry for this. We have updated the reference list, so as to reflect the real number of citations. Thank you